# Adaptation and Psychometric Properties of the Behavioral Regulation in Exercise Questionnaire (BREQ-3) for Motivation Towards Incidental Physical Activity

**DOI:** 10.3390/bs15020114

**Published:** 2025-01-23

**Authors:** Daniel Reyes-Molina, Rafael Zapata-Lamana, Claudio Bustos, Javier Mella-Norambuena, Isidora Zañartu, Yasna Chávez-Castillo, Jorge Gajardo-Aguayo, Anabel Castillo-Carreño, María-Francisca Cabezas, Víctor Castillo Riquelme, Tomás Reyes-Amigo, Igor Cigarroa, Gabriela Nazar

**Affiliations:** 1Doctorado en Psicología, Facultad de Ciencias Sociales, Universidad de Concepción, Concepción 4030000, Chile; danielreyes@udec.cl (D.R.-M.); zanartu@udec.cl (I.Z.); yasnasolchavez@udec.cl (Y.C.-C.); jorgegajardo@udec.cl (J.G.-A.); anabelicastillo@udec.cl (A.C.-C.); 2Escuela de Kinesiología, Facultad de Salud, Universidad Santo Tomás, Los Ángeles 4430000, Chile; rafaelzapata@udec.cl; 3Escuela de Educación, Universidad de Concepción, Los Ángeles 4440000, Chile; 4Departamento de Psicología, Universidad de Concepción, Concepción 4030000, Chile; clbustos@udec.cl; 5Departamento de Ciencias, Universidad Técnica Federico Santa María, Concepción 4030000, Chile; javier.mellan@usm.cl; 6Departamento Fundamentos de Enfermería y Salud Pública, Facultad de Enfermería, Universidad de Concepción, Concepción 4030000, Chile; 7University of Groningen, University Medical Center Groningen, Department of Health Sciences, Section of Health Psychology, Hanzeplein 1, 9713 G, 9713 GZ Groningen, The Netherlands; m.f.cabezas.henriquez@umcg.nl; 8Escuela de Psicología, Universidad Santo Tomás, Los Ángeles 4430000, Chile; vcastillo10@santotomas.cl; 9Observatorio de Ciencias de Ciencias de la Actividad Física (OCAF), Departamento de Ciencias de la Actividad Física, Universidad de Playa Ancha, Valparaíso 2340000, Chile; tomas.reyes@upla.cl; 10Escuela de Kinesiología, Facultad de Ciencias de la Salud, Universidad Católica Silva Henríquez, Santiago 8330225, Chile; icigarroac@ucsh.cl; 11Centro de Vida Saludable, Universidad de Concepción, Concepción 4030000, Chile

**Keywords:** BREQ-3, psychometric properties, factor analysis, reliability, incidental physical activity, self-determination theory

## Abstract

This study aimed to adapt and analyze the psychometric properties of the Exercise Behavior Regulation Questionnaire (BREQ-3) for assessing motivation towards incidental physical activity. An instrumental study in a sample of 346 university students (21.1 ± 2.6 years, and 61.3% women) from various universities in Chile was undertaken. An adaptation of the BREQ-3 was applied, and a confirmatory factor analysis was performed using a robust weighted least squares estimator to assess the construct validity of the scale. Also, the convergent validity was evaluated using the average variance extracted, the discriminant validity using composite reliability, and the internal consistency using Cronbach’s alpha (α) and McDonald’s omega (ω) coefficients. The six-factor structure of intrinsic motivation (α = 0.96, ω = 0.96), integrated regulation (α = 0.95, ω = 0.95), identified regulation (α = 0.89, ω = 0.90), introjected regulation (α = 0.75, ω = 0.77), external regulation (α = 0.80, ω = 0.83), and amotivation (α = 0.75, ω = 0.79), with acceptable fit indices after eliminating items 8 and 11, was confirmed—χ^2^/df: 2.196, CFI: 0.99, TLI: 0.99, RMSEA: 0.059 (90% CI; 0.051–0.067). Adaptation of the BREQ-3 appears to be a reliable measure for assessing motivation in the context of incidental physical activity. Its use will contribute to understanding the explanatory mechanisms underlying this behavior.

## 1. Introduction

Currently, approximately 31.3% of the adult population worldwide and 32.3% in Chile ([78]) do not meet the physical activity recommendations of the World Health Organization ([7]). The benefits of increasing physical activity include improved health and well-being ([20]; [21]; [42]; [43]; [88]). So, promoting physical activity is a public health priority ([76]; [85]).

In this sense, understanding the factors that influence and predict physical activity behavior is of broad interest to the scientific community. In particular, motivational factors have been widely addressed ([64]; [38]; [54]; [80]) using the self-determination theory as one of the theoretical references ([18]).

### 1.1. Theory of Self-Determination

Self-determination theory has been used to explain a wide variety of human behaviors in different contexts, including work ([25]; [37]), education ([53]; [66]; [79]), health ([26]; [60]), sports ([10]; [31]; [77]), and also for physical exercise ([30]; [35]).

This theory points out that motivation is seen as a continuum of self-determination. At one end of this continuum is amotivation or demotivation, understood as the absence of intention towards a specific behavior. At the other extreme is intrinsic motivation, where an activity is performed because there is a natural incentive to act due to the internal satisfaction derived from that behavior. In intrinsic motivation, activity is pleasurable in itself, and its interest and enjoyment are inherent ([18], [19]; [70]).

A third form of motivation, extrinsic motivation, lies between amotivation and intrinsic motivation. Extrinsic motivation involves engaging in an activity to obtain some outcome separate from the activity itself, as it presents instrumentally manipulated and goal-oriented incentives to carry out the behavior. Four forms of extrinsic motivation, called regulations, are given below in order of the least to the highest degree of self-determination: external regulation, introjected regulation, identified regulation, and integrated regulation ([18]; [70]).

External regulation refers to behaviors driven by external rewards. Introjected regulation is a partially internalized form of self-regulation where people engage in activities motivated by internal rewards, punishments, self-esteem, or ego. In identified regulation, people consciously identify with the value of an activity and, therefore, experience a high degree of desire, willingness, or willingness to act. In integrated regulation, people not only recognize and identify the value of the activity but also find it to be congruent with their interests and values ([70]).

Based on the self-determination theory, instruments have been developed that allow the theoretical evaluation of the relationship between motivation and different types of physical activity, such as physical exercise or sport. These instruments have been updated in more comprehensive versions over the years and have been translated into several languages to expand their use. These include the Perceived Locus of Causation Questionnaire ([69]), the Behavioral Regulation in Sport Questionnaire ([48]), and the Exercise Behavioral Regulation Questionnaire (BREQ) ([58]).

### 1.2. Behavioral Regulation in Exercise Questionnaire (BREQ)

The Behavioral Regulation in Exercise Questionnaire (BREQ) was developed by [58] ([58]) to assess levels of motivation towards physical exercise based on self-determination theory ([58]). In its first version, it opted for a 15-item model grouped into four factors with adequate model fit indices (χ^2^ = 277.19, *p* < 0.001, root mean square error of approximation or RMSEA = 0.07) and internal consistency using Cronbach’s alpha (external regulation = 0.79, introjected regulation = 0.78, identified regulation = 0.79, intrinsic motivation = 0.90). This first version did not consider the dimension of amotivation because it was statistically problematic and left out integrated regulation ([58]). In this way, the model consisted of the dimensions of external regulation, introjected regulation, identified regulation, and intrinsic motivation ([58]). Although some evidence of validity was observed for this first version of the BREQ, the exclusion of the dimension of amotivation remained incongruent with the taxonomy of motivation based on self-determination theory.

Subsequently, a second version of the BREQ (BREQ-2) developed by [55] ([55]) of 19 items grouped into five factors incorporated the dimension of amotivation. The BREQ-2 presented an excellent fit of the model to the data (χ^2^ (125) = 136.49, *p* = 0.23, RMSEA = 0.02, comparative fit index or CFI = 0.95, Tucker–Lewis indices or TLI = 0.94, standardized root mean square residual or SRMR = 0.05), and also an internal consistency at acceptable levels ([55]). The original BREQ-2 demonstrated satisfactory evidence of construct validity in English for university students ([17]), and also for its translations in different languages in the general population ([15]; [46]; [57]).

A third version of the questionnaire (BREQ-3) was developed to resolve the conceptual discrepancy between self-determination theory and BREQ by including integrated regulation. However, this new version initially left out the dimension of amotivation ([84]). It was then that [28] ([28]) expanded on the work done by [84] ([84]) and by [55] ([55]), developing the BREQ-3 in Spanish with the inclusion of the dimension of motivation. In this way, the BREQ-3 in Spanish comprised 23 items grouped into six dimensions that covered the continuum of self-determination from amotivation to intrinsic motivation ([28]). The BREQ-3 in Spanish presented adequate fit indicators—χ^2^ (215) = 689.13, *p* < 0.001; χ^2^/df = 3.20; CFI = 0.91; RMSEA = 0.06; SRMR = 0.06 ([28]). The BREQ-3 has been adapted and validated in countries such as Germany, Brazil, China, and Italy, where it has presented acceptable adjustment indices and invariance across gender and age ([9]; [12]; [14]; [51]). The BREQ-3 has evidence of application in the adult population in Chile ([40]; [56]; [67]), having an adequate internal consistency for university students ([56]).

### 1.3. BREQ-3 and Incidental Physical Activity

BREQ was designed for use in physical exercise, that is, a planned, structured, repetitive physical activity that aims to improve physical fitness ([8]; [22]). However, there are other types of less structured physical activities, such as incidental physical activity, which have growing evidence of a positive association with indicators of physical health and well-being ([1]; [65]; [73], [72]; [87]) in addition to a higher adherence rate over time than physical exercise ([11]; [44]; [52]). Incidental physical activity is characterized as physical activity of daily living, performed at home, at work or study, or during leisure time, that does not require additional time or have recreational or physical health purposes (e.g., climbing stairs, active transportation, standing, shopping, household chores, occupational, physical activity, among others.) ([68]; [75], [74]).

Adaptations of motivational instruments have been made for the study of incidental physical activity ([59]; [62]). In particular, Oliver and Kemps adapted the Perceived Locus of Causality Questionnaire ([69]) and found that both more self-determined motivations and introjected and external regulations predicted levels of incidental physical activity in a group of college students. Another study adapted the BREQ-3 to analyze motivation in walking behavior, which is considered a sub-behavior of incidental physical activity. The results showed negative relationships with amotivation and external regulation, no relationship with introjection, and positive relationships with identified and intrinsic regulation ([59]). However, according to the empirical review carried out, there are no published studies that have adapted the BREQ-3 to measure motivation towards incidental physical activity. Having an instrument such as the BREQ-3 to assess incidental physical activity would allow us to understand the motivational factors underlying this type of physical activity, as well as to generate scientific evidence from various populations from different cultural backgrounds.

With the above background, this study aimed to adapt and analyze the psychometric properties of the Exercise Behavior Regulation Questionnaire (BREQ-3) for assessing motivation towards incidental physical activity.

## 2. Materials and Methods

### 2.1. Design

A quantitative approach was followed with an instrumental study design ([3]) developed from the cross-sectional survey method ([16]) through collecting information at a single point in time. This study followed the recommendations for validating psychometric instruments in social and health sciences ([49]).

### 2.2. Participants

Through a non-probabilistic convenience sampling, 346 university students were recruited, aged between 18 and 34 years (mean 21.1 ± 2.6 years; 61.3% women), from 11 universities in different regions of Chile. Of these, 54% (n = 188) practiced sports or physical exercise at least once a week. The inclusion criterion was to be a regular undergraduate student in a higher education institution. No exclusion criteria related to physical exercise, level of physical activity, or other factors were applied since the study focused on motivation toward incidental physical activity. This type of activity corresponds to activities of daily living carried out by both people who practice physical exercise and those who do not.

All participants were informed about the general purpose of the study and their rights to anonymity and confidentiality. They agreed to participate in the study by signing informed consent forms before answering the questionnaire.

### 2.3. Instruments

The third version of the Exercise Behavior Regulation Questionnaire was adapted to Spanish (BREQ-3) ([28]; [84]). The BREQ-3, in its Spanish version, is composed of 23 items grouped in six dimensions, which evaluate amotivation, external regulation, introjected, identified, integrated, and intrinsic motivation ([28]; [84]). All items begin with the statement “I do physical exercise” and are rated on a five-point Likert-type scale, ranging from “0” (not true at all) to “4” (totally true). Items 4, 12, 18, and 22 correspond to the dimension of intrinsic motivation (example item, “because I think exercise is fun”); items 5, 10, 15, and 20 for integrated regulation (example item, “because it is in accordance with my way of life”); items 3, 9, and 17 for identified regulation (example item, “because I value the benefits of physical exercise”); items 2, 8, 16, and 21 for introjected regulation (example item, “because I feel guilty when I do not practice it”); items 1, 7, 13, and 19 for external regulation (example item, “because others tell me I should do it”); and items 6, 11, 14, and 23 for amotivation (example item, “I do not see why I have to”). The BREQ-3 scale, initially in English, has been adapted and validated in countries such as Brazil, Italy, and Spain, with psychometric properties that indicate acceptable fit indices and that corroborate its six-dimensional factor structure and with invariance across gender and age ([9]; [12]; [28]). In university students in Chile, internal consistency was reported with Cronbach’s alpha values of α = 0.87 for intrinsic motivation, α = 0.87 for integrated regulation, α = 0.66 for identified regulation, α = 0.72 for introjected regulation, α = 0.78 for external regulation, and α = 0.70 for amotivation ([56]).

### 2.4. Procedure

First, permission was requested from the corresponding author of the BREQ-3 in its Spanish version to adapt the instrument, and who gave his consent for its free use. Next, the adaptation process began, which included adding a statement to the instructions of the original questionnaire, with the purpose of placing the university students in the context of the incidental physical activities of daily life, which they had to keep in mind when answering the questionnaire (see Appendix A). Then, the statement “*I do physical activity*” was prefixed to each item, unlike the original questionnaire, where this was only in the instructions. The concept of “*physical exercise*” was replaced throughout the questionnaire by “*physical activity*”.

In order to evaluate the adaptations made to the original questionnaire, to verify the correct understanding of the items, and for the relevance of the questionnaire in the context of university students, cognitive interviews were carried out ([71]). The interviews used paraphrasing techniques, in which the interviewer rephrased the question to verify comprehension and clarify meaning, together with specification tests using additional questions, such as how they defined or interpreted certain terms, to explore how the participant arrived at their answer. All of the above had the purpose of detecting problems in the four moments of the question–answer process (comprehension, information retrieval, estimation, and execution) ([71]). Participants for cognitive interviews were selected through convenience sampling among students who met the same inclusion criteria as the study and who also signed the informed consent ([63]). Eight students from a university in the Biobío region participated, three men and five women, between 18 and 27 years old, from engineering and pedagogy careers. Each interview was audio-recorded, lasted approximately 60 min, and took place at a pre-arranged location. Each student was provided with a laptop so that they could read aloud the questionnaire available on the Google Forms platform while the interviewer (principal investigator) recorded, asked questions, and took notes. The answers submitted by the students to each of the items were analyzed, and the necessary modifications were made where comprehension difficulties were identified. Five out of the eight students expressed doubts or suggestions about one of the 23 items on the questionnaire. This led to light modifications being made in the wording of items 2, 3, 6, 8, 11, 12, 13, 14, 16, 18, and 23 (see Table 1).

After these modifications, the online questionnaire was applied to university students from all over Chile through the snowball method. The link to access the questionnaire hosted on the Google Forms platform was open access, and the research team disseminated it through social networks and emails between November 2023 and June 2024.

### 2.5. Data Analysis

The first step was to carry out a qualitative analysis through a thematic analysis of the results obtained in the cognitive interviews. The next step, before proceeding with factor analyses, was to review the suitability of the data for analysis. A descriptive analysis of the data showed that there were no missing data. The mean and standard deviation were presented for each item. Then, a confirmatory factor analysis was carried out on the sample of 346 participants. All of this was estimated with the support of the statistical tool R (version 4.4.1), using the packages “psych” and “lavaan”. Since the study variables were measured with ordinal scales, a robust weighted least squares estimator (WLSMV) was used ([5]). To evaluate the fit of the models, chi-square indices (χ^2^; expected values *p* > 0.05) were used; the mean square root of approximation error (RMSEA) (which is considered a good fit if it has values less than 0.05–0.08); and the comparative fit index (CFI) and Tucker–Lewis indices (TLI) (where values > 0.95 are expected) ([5]; [34]) The convergent and discriminant validity of the factors was analyzed using the average variance extracted (AVE) and the composite reliability (CR). The AVE measures the level of variance captured by a construct compared to the variance attributed to measurement error. According to [24] ([24]), discriminant validity compares the square root of the AVE with the correlations between latent constructs. A latent construct should explain more variance in itself than the variance shared with other latent constructs. Therefore, the square root of each AVE should be greater than the correlations between the other latent constructs. The recommended cut-off points are 0.50 for the AVE and 0.70 for the CR ([24]; [34]). Subsequently, Cronbach’s alpha and McDonald’s omega coefficients were calculated to evaluate the internal consistency of the constructs, where values > 0.70 are expected ([50]).

## 3. Results

Table 2 presents the descriptive statistics for each item and the dimensions assessed in the instrument. The mean, standard deviation (SD), and minimum and maximum values observed (min–max) are reported on a scale from 0 to 4 for each dimension. The item means per dimension show a trend toward higher levels of intrinsic motivation and identified regulation, while external regulation and amotivation are the least predominant.

### 3.1. Confirmatory Factor Analysis

The original model of the BREQ-3 questionnaire, which consists of six factors, was evaluated. Then, a second model was evaluated, excluding items 8 and 11, because the residue analysis showed a high residual relationship between these two items (>0.05). Both models presented a significant chi-square (χ^2^), which indicates a significant difference between the expected and observed covariance matrix, suggesting that the models proposed in both cases do not correctly represent the relationships between the observed variables. However, the CFI and TLI values are appropriate. Model 2 also presented a χ^2^/df and RMSEA in adequate ranges, with values lower than three and 0.08, respectively, which indicates a better fit (see Table 3).

### 3.2. Convergent and Discriminant Validity

The convergent and discriminant validity indicators show that in both models, the average variance extracted (AVE) is adequate for the factors of intrinsic motivation and integrated and identified regulation since it exceeds the threshold of 0.50. However, amotivation has a PAVE lower than 0.50 in Model 1 (AVE = 0.35). The modification in Model 2, which excludes items 8 and 11, improves the AVE of amotivation (AVE = 0.51), reaching an adequate value. The composite reliability (CR) in both models is acceptable for all factors, except for the amotivation factor in Model 1, with a CR lower than 0.7 (CR = 0.67), which, after modification in Model 2, improves to acceptable values (CR = 0.75) (see Table 4).

### 3.3. Internal Consistency Analysis

The internal consistency in both models presents adequate values of Cronbach’s alpha (α) and McDonald’s omega (ω) for the factors of intrinsic motivation and integrated and identified regulation, with values above 0.90. In Model 2, the internal consistency indicators for the amotivation factor improve compared to Model 1, reaching an α = 0.75 and ω = 0.79, respectively (see Table 4).

The six-factor Model 2 was chosen without items 8 and 11 since it shows a robust fit, with good indices of χ^2^/df, CFI, TLI, and RMSEA. In addition, the model showed adequate convergent and discriminant validity, with AVE and CR in acceptable values for all its dimensions. Internal consistency was also strong, with α and ω values ranging from good (>0.60) to excellent (>0.90). All items had factor loads greater than 0.60, and most exceeded 0.80, except for item 1 (0.75), item 2 (0.63), and item 6 (0.73). Intrinsic motivation, integrated regulation, and identified regulation showed high correlation with each other, with values equal to or greater than 0.90, as well as a low correlation with the introjected regulation factor, and a low and negative correlation with external regulation and amotivation (see Figure 1).

## 4. Discussion

This research aimed to adapt and analyze the BREQ-3 psychometric properties for the assessment of motivation towards incidental physical activity. Confirmatory factor analysis supported the six-factor structure corresponding to the dimensions of intrinsic motivation, integrated regulation, identified regulation, introjected regulation, external regulation, and amotivation of the original BREQ-3 model in its Spanish version ([28]). These findings are in line with validation studies and psychometric properties of BREQ-3 in other languages and population groups, where the six-factor model has been shown to be a good fit for the data ([9]; [12]; [28]; [29]; [51]).

The adjustment indices of the adaptation of the BREQ-3 that we proposed to study the motivation towards incidental physical activity, after eliminating items 8 and 11, were similar to those of the Spanish version of the BREQ-3 by [28] ([28]) —χ^2^ = 215, *p* < 0.001, χ^2^/df = 3.20; CFI = 0.91; TFI = 0.91; RMSEA = 0.06. It is important to note that the version of the BREQ-3 by [28] ([28]) was applied to a sample composed of practitioners from various exercise contexts, while in the present study, the sample consisted of university students, both those who practiced sports or exercised (54%) and those who did not. In this sense, the original version of the BREQ-3 was designed to assess motivation towards physical exercise ([28]; [84]), although it may share types of motivation with incidental physical activity ([45]; [61]), which refer to activities with a different purpose and context. This could explain some of the differences found in the initial fit of the model in the adaptation we proposed, especially with item 8 (“I do the physical activity because I feel embarrassed if I do not do it”), belonging to the dimension of amotivation, and item 11 (“I do physical activity, but I do not see why I have to bother doing it”) of the introjected regulation dimension. These items would not fit the context of incidental physical activity, an activity that is part of daily life and is reflected in behaviors such as walking, climbing stairs, doing the toilet, or shopping, among others. In this sense, the improvement in the convergent and discriminant validity indices (AVE and CR) for the dimension of amotivation, after eliminating the items, confirms this observation.

On the other hand, the correlations between the dimensions of the instrument were consistent with the theory, presenting the most self-determined dimensions, such as intrinsic motivation and integrated and identified regulation, with high correlations with each other. In addition, these more self-determined dimensions showed negative correlations with less self-determined dimensions, such as introjected and external regulation, as well as with amotivation. This pattern of correlations is consistent in the various linguistic adaptations of the BREQ-3 identified in a recent systematic review ([82]), which supports the multidimensionality of motivation as a continuum of self-determination ([2]; [4]; [51]).

Notably, although the classical six-dimensional taxonomy of motivation is widely used, an alternative classification has been proposed that groups the self-determination continuum into three categories: amotivation, controlled motivation (which includes introjected and external regulation), and autonomous motivation (which includes intrinsic motivation and integrated and identified regulation) ([83]). In particular, a study of 825 university students in China showed acceptable fit indices—χ^2^ (431) = 808.070, *p* < 0.01; RMSEA = 0.053 (0.047–0.059); CFI = 0.93; TLI = 0.93—and an adequate internal consistency for a three-factor model, in which the items were grouped into the dimensions of amotivation, controlled motivation, and autonomous motivation ([51]) This taxonomy, observed in the research that applied the BREQ-3, could explain the correlations between the dimensions observed in the present study. Other proposals have integrated behavioral regulations as second-order variables in models regrouped into two factors: autonomous and controlled motivation ([6]; [13]; [39]; [51]). While these measures of motivation have been consistent with the central distinction underlying the self-determination continuum proposed in self-determination theory ([13]), combining the subscales could dismiss the differential consequences associated with each regulation ([39]).

### Strengths and Limitations

To our knowledge, this is the first study to explore the psychometric properties of an adaptation of the BREQ-3 for the analysis of motivation towards incidental physical activity. This study reported on the construct, convergent, discriminant validity, and internal consistency of this adaptation of the BREQ-3, making it a reliable and psychometrically suitable scale for use in the context of incidental physical activity. Understanding the different forms of motivation that people experience during exercise can help predict potential outcomes ([33]). This study contributes significantly to the literature by expanding the instruments available for motivational research, which may facilitate international comparisons.

However, the present study has limitations that must be considered when interpreting its results. First, although sample sizes of more than 200 cases are considered adequate to assess the quality of a test ([23]), the optimal conditions suggest having more than 400 cases, which allow exploratory and confirmatory factor analyses to be carried out ([47]). Second, the study did not include a measure of incidental physical activity or a re-test in the sample of students to whom the instrument was applied. Therefore, in the present study, there are no indicators of concurrent validity or temporal stability of the adaptation of the BREQ-3. Finally, the version of the BREQ-3 validated in Spanish that was used in this study has 23 items, unlike the English versions of the BREQ-3 adapted to other languages, such as German, Portuguese, and Chinese, which include an additional item in the identified regulation dimension, adding up to a total of 24 items, with four items per factor ([12]; [14]; [51]).

Future research should evaluate the psychometric properties of this adaptation of BREQ-3 in larger samples and in populations of various ages and occupations. Its concurrent validity to objective measures of incidental physical activity and the temporal stability of the instrument should also be examined, as well as to determine if this new version of the BREQ-3 maintains the original six-factor structure or if it fits into a three-factor structure corresponding to the dimensions of autonomous motivation, controlled motivation, and amotivation. It would also be desirable to study the psychometric properties of this adaptation, including the fourth item in the identified regulation dimension that was incorporated into the English versions of BREQ-3.

This adaptation of the BREQ-3 will contribute to the research on motivation towards incidental physical activity, where evidence on the psychological mechanisms that explain these behaviors is limited ([44]; [68]; [81]). To date, evidence on the psychological factors influencing incidental physical activity has focused primarily on the interaction between the continuum of motivation and implicit processes, such as attitudes, attentional biases, and approach-avoidance mechanisms ([62]). However, there is literature that describes, for example, how the incorporation of the self-determination continuum into the theory of planned behavior can help explain behaviors related to physical activity ([27]; [32]). It has even been described how the most self-determined motivations mediate between personality traits, the constructs of the theory of planned behavior, and physical activity ([41]).

## 5. Conclusions

The present study, which adapted and analyzed the psychometric properties of BREQ-3, provides evidence that it is an adequate measure to assess motivation in the context of incidental physical activity. The confirmatory factor analysis showed a good fit to the six-factor model, supporting the structure corresponding to the dimensions of intrinsic motivation, integrated regulation, identified regulation, introjected regulation, external regulation, and amotivation. Future research should address the limitations of this research, such as sample size and lack of measures of concurrent validity and temporal stability. However, this adaptation of the BREQ-3 offers a valid and reliable tool in the study of the psychological mechanisms underlying motivation towards incidental physical activity, an area that has so far received little attention.

## Figures and Tables

**Figure 1 behavsci-15-00114-f001:**
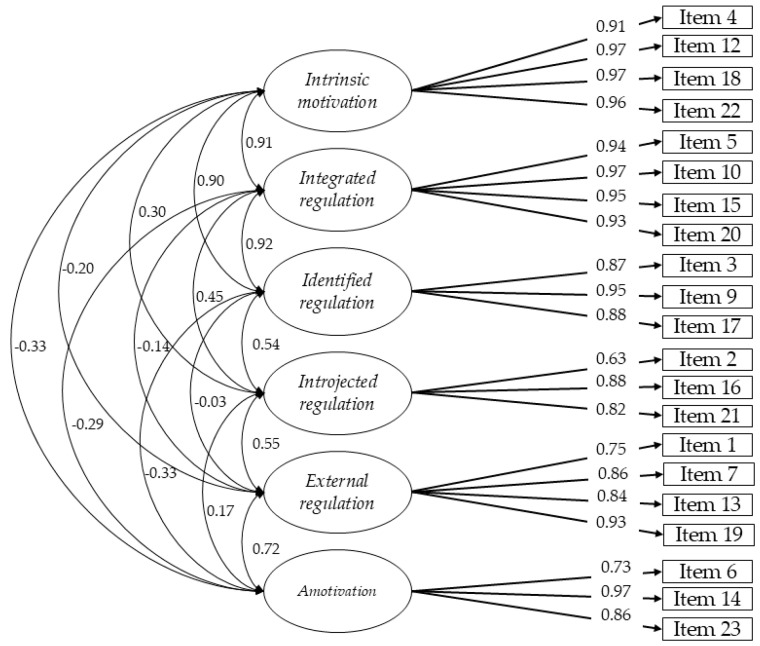
Diagram of the multifactorial structure of the version of the BREQ-3 for motivation towards incidental physical activity.

**Table 1 behavsci-15-00114-t001:** Modifications were made after the cognitive interview.

Item	Original Questionnaire	Modification
2	Because I feel guilty when I do not practice it.	I do physical activity because I feel guilty when I do not.
3	Because I value the benefits of physical exercise.	I do physical activity because I value the benefits of doing it.
6	I do not see why I have to.	I do physical activity, but I do not see why I have to do it.
8	Because I feel embarrassed if I miss the session.	I do physical activity, but I do not see why I have to do it.
11	I do not see why I have to bother exercising.	I do physical activity, but I do not see why I have to bother doing it.
12	Because I enjoy exercise sessions.	I do physical activity because I enjoy doing it.
13	Because other people will not be happy with me if I do not exercise.	I do physical activity because other people will not be happy with me if I do not do physical activity.
14	I do not see the point of exercising.	I do physical activity, but I do not see the point of doing physical activity.
16	Because I feel like I have failed when I have not exercised for a while.	I do physical activity because I feel that I have failed when I have not done physical activity for a while.
18	Because I find exercise an enjoyable activity.	I do physical activity because I find it to be an enjoyable activity.
23	I think exercising is a waste of time.	I do physical activity, but I think that doing it is a waste of time.

**Table 2 behavsci-15-00114-t002:** Descriptive statistics by item and by dimension.

Dimension	Item	Mean (SD)	Min–Max
Intrinsic motivation	4	2.52 ± 1.50	0–4
	12	2.49 ± 1.52	0–4
	18	2.54 ± 1.49	0–4
	22	2.42 ± 1.51	0–4
Integrated regulation	5	2.25 ± 1.53	0–4
	10	1.96 ± 1.62	0–4
	15	1.91 ± 1.65	0–4
	20	1.95 ± 1.55	0–4
Identified regulation	3	2.65 ± 1.39	0–4
	9	2.41 ± 1.50	0–4
	17	2.36 ± 1.46	0–4
Introjected regulation	2	1.14 ± 1.29	0–4
	8	0.57 ± 1.03	0–4
	16	1.11 ± 1.35	0–4
	21	0.84 ± 1.22	0–4
External regulation	1	0.47 ± 0.89	0–4
	7	0.39 ± 0.84	0–4
	13	0.28 ± 0.75	0–4
	19	0.27 ± 0.69	0–4
Amotivation	6	0.38 ± 0.86	0–4
	11	0.75 ± 1.18	0–4
	14	0.25 ± 0.71	0–4
	23	0.23 ± 0.67	0–4

SD: standard deviation.

**Table 3 behavsci-15-00114-t003:** Adjustment of the models of the Regulation of Behavior in Exercise Questionnaire (BREQ-3) analyzed.

Model	χ^2^	χ^2^/df	CFI	TLI	RMSEA
Model 1: Six factors	781.58 *	3.635	0.99	0.99	0.087 (IC 90%; 0.081–0.094)
Model 2: Six factors without items 8 and 11	382.18 *	2.196	0.99	0.99	0.059 (IC 90%; 0.051–0.067)

χ^2^: chi-square; df: degree of freedom; CFI: comparative adjustment index; TLI: Tucker–Lewis index; RMSEA: mean square root of approximation error. * *p*-value < 0.001.

**Table 4 behavsci-15-00114-t004:** Indicators of convergent and discriminant validity and internal consistency.

Model		AVE	CR	α	ω
Model 1	Intrinsic motivation	0.86	0.96	0.96	0.96
	Integrated regulation	0.84	0.95	0.95	0.95
	Identified regulation	0.75	0.90	0.89	0.90
	Introjected regulation	0.50	0.80	0.79	0.82
	External regulation	0.51	0.81	0.80	0.83
	Amotivation	0.35	0.67	0.68	0.73
Model 2	Intrinsic motivation	0.86	0.96	0.96	0.96
	Integrated regulation	0.84	0.95	0.95	0.95
	Identified regulation	0.75	0.90	0.89	0.90
	Introjected regulation	0.52	0.76	0.75	0.77
	External regulation	0.51	0.80	0.80	0.83
	Amotivation	0.51	0.75	0.75	0.79

AVE: average extracted variance; CR: composite reliability; α: Cronbach’s alpha; ω: McDonald’s omega.

## Data Availability

The data presented in this study are available on request from the corresponding author due to restrictions imposed to safeguard the identity of the subjects studied and to comply with applicable ethical regulations.

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
