# Peer review of "Adaptation and Psychometric Properties of the Behavioral Regulation in Exercise Questionnaire (BREQ-3) for Motivation Towards Incidental Physical Activity"

_behavsci, 2025, doi:10.3390/bs15020114_

Round 1
Reviewer 1 Report
Comments and Suggestions for Authors
Perhaps the abstract could be adapted to the requirements of the journal template.
Author Response
Comment 1: Perhaps the abstract could be adapted to the requirements of the journal template.
Response 1: We appreciate the reviewer's comment. In this regard, we have verified again that our abstract meets the journal's requirements. In this regard, we would like to confirm that we have presented a single-paragraph abstract within the word limit, which includes information on the objective, method, results, and conclusions.
Reviewer 2 Report
Comments and Suggestions for Authors
Introduction
The introduction provides a generally strong argumentation.
In my opinion, it could be improved by:
- Including psychometric data for all versions of the BREQ and BREQ-3.
Method
The sample selection is intentional. Although this might not pose a problem for validation purposes, it could limit the generalizability of the results to the Chilean population. It is recommended to include information on:
- The percentage of men and women in Chile.
- The percentage of people who engage in sports and physical exercise in Chile.
These data would allow for a better assessment of the sample's representativeness concerning its population of origin.
Regarding the process of verifying item comprehension conducted with 8 students, it would be highly valuable to report how many participants did not understand the items and to what extent there was agreement among the others. Using a concordance statistic between observers could provide additional insights.
Furthermore, it is necessary to clarify how it was ensured that participants in the validation process understood the meaning of "physical activity." In non-specialized populations, there is likely a lack of understanding of the distinction between "physical exercise" and "physical activity." Was there any explanation of these terms included at the beginning of the questionnaire?
Distributing the questionnaire through social media could represent a limitation of the study, as it does not control for who actually completed the form. The researchers should specify the methods used to address this potential confounding variable (e.g., registration via email, verification processes to avoid bots, etc.).
Results
The article would be more engaging for readers if it included descriptive results of the values obtained by participants on the scales. I understand that the authors might be planning a separate article, but given that these results are already available at the time of writing this study, I believe it is important to include them here.
To incorporate these results, it would be essential to detail in the methods section the statistical measures to be used.
Author Response
Comment 1: Introduction. The introduction provides a generally strong argumentation.
In my opinion, it could be improved by: Including psychometric data for all versions of the BREQ and BREQ-3.
Response 1: Based on the reviewer's comment, we have improved the background information quality by incorporating the BREQ versions' psychometric properties.
In this regard, lines 93 to 95 now read: With indicators of model fit (χ² = 277.19, p < .001, RMSEA = .07) and adequate Cronbach's alpha (external regulation = 0.79, introjected regulation = .78, identified regulation = .79, intrinsic motivation = .90).
And in the lines 119 to 121 now read: The BREQ-3 in Spanish presented adequate fit indicators χ2 (215)= 689.13, p < .001; χ2/gl= 3.20; CFI= .91; RMSEA= .06; SRMR= .06 (González-Cutre et al., 2010).
Comment 2: Method. The sample selection is intentional. Although this might not pose a problem for validation purposes, it could limit the generalizability of the results to the Chilean population. It is recommended to include information on: The percentage of men and women in Chile. And The percentage of people who engage in sports and physical exercise in Chile. These data would allow for a better assessment of the sample's representativeness concerning its population of origin.
Response 2: We appreciate the reviewer's valuable comment and fully understand the point made. However, we wish to clarify that this study was not intended to be representative of the Chilean population or of university students in Chile. The sample size was determined based on methodological requirements, specifically those associated with confirmatory factor analysis. Therefore, we consider that including information such as the percentage of men and women or the percentage of people who practice sports and physical exercise could lead to incorrect interpretations, suggesting a purpose that was not contemplated in this study.
Comment 3: Regarding the process of verifying item comprehension conducted with 8 students, it would be highly valuable to report how many participants did not understand the items and to what extent there was agreement among the others. Using a concordance statistic between observers could provide additional insights.
Response 3: We appreciate the reviewer's comment. In this regard, we have included information on the number of participants who reported difficulties in understanding some items in the questionnaire in the method procedure section of our manuscript. Thus, between lines 231 and 234, it now reads: Five out of the eight students expressed doubts or suggestions about one of the 23 items on the questionnaire. This led to modifications being made in the wording of items 2, 3, 6, 8, 11, 12, 13, 14, 16, 18, and 23 (see Table 1).
We would also like to let you know that Table 1, which was previously located at the beginning of the results section, has been moved to the methodology procedure section. This change makes it easier to view the information provided in the procedure paragraph and table regarding the modifications made to the items. In this way, the results section is left exclusively to report the findings from the analyses. It now also includes descriptive information about the sample and the data distribution.
Comment 4: Furthermore, it is necessary to clarify how it was ensured that participants in the validation process understood the meaning of "physical activity." In non-specialized populations, there is likely a lack of understanding of the distinction between "physical exercise" and "physical activity." Was there any explanation of these terms included at the beginning of the questionnaire?
Response 4: To ensure that participants were clear about the physical activity being referred to, a paragraph was used at the beginning of the questionnaire, before the instructions (see Supplementary Table) describing the type of physical activity being referred to in each item. The text was as follows:
To answer the following questionnaire, consider only those physical activities performed in daily life, i.e., those performed at home, at study or work, and in free time, that are not for health or fitness purposes. For example, activities such as using stairs, walking or cycling, gardening or housework, activities during school or work hours, walking the dog or pets, shopping, and children's play activities, among others. When answering the questionnaire, physical exercise performed in gyms, sports workshops, or sports at a competitive or recreational level should not be considered.
Comment 5: Distributing the questionnaire through social media could represent a limitation of the study, as it does not control for who actually completed the form. The researchers should specify the methods used to address this potential confounding variable (e.g., registration via email, verification processes to avoid bots, etc.).
Response 5: We appreciate the reviewer's comment. To ensure that participants were real people, we implemented an email registration as a requirement to access the questionnaire. While we cannot be 100% sure that all participants were university students, we consider this to be highly probable, given that (1) Sociodemographic questions related to the major, year of study, and other aspects linked to the university environment were included. (2) Most participants accessed the questionnaire through links provided by teachers, who distributed it exclusively among their students. And (3) the questionnaire was part of a broader research that included other batteries and self-report questionnaires, whose length and time required to complete them could have discouraged the participation of people outside the university context.
Comment 6: Results. The article would be more engaging for readers if it included descriptive results of the values obtained by participants on the scales. I understand that the authors might be planning a separate article, but given that these results are already available at the time of writing this study, I believe it is important to include them here. To incorporate these results, it would be essential to detail in the methods section the statistical measures to be used.
Response 6: We appreciate the reviewer's comment, which has allowed us to improve the quality of the findings reported in the results section. In this regard, we have incorporated a new table (Table 2) in which the means and standard deviations obtained by item are reported, according to each dimension. In addition, this information has been detailed in the statistical analysis section of the methodology. Now, between lines 245 and 256, this information is included: The mean and standard deviation were presented for each item.
Reviewer 3 Report
Comments and Suggestions for Authors
I hope this letter finds you well. I had the opportunity to review your article titled, “Adaptation and psychometric properties of the Behavioral Regulation in Exercise Questionnaire (BREQ-3) for motivation towards incidental physical activity”, which was submitted Behavioral Sciences.
1. Abstract
Please provide a brief explanation of the research methods.
2. Introduction
When presenting previous studies in the text, place citations at the end of sentences.
Each paragraph should consist of at least three sentences.
The verbs in the theoretical background should be written in the past tense.
There is a tendency for sentences to become lengthy, leading to ambiguity.
Write concise sentences to ensure the author's arguments are conveyed accurately.
This study's purpose of exploring the applicability of BREQ is considered excellent.
In particular, survey instruments need continuous validation and reliability testing over time.
From this perspective, this study is highly interesting and necessary in the field.
3. Materials and Methods
[63] Adjust the font size appropriately.
2.2 Participants
The study participants consist solely of women.
If so, the title should specify that the study is limited to women.
Additionally, explain why the study focuses specifically on college students.
2.3 Instruments and 2.4 Procedure
These sections are very well written.
Ensure uniformity in all mathematical symbols.
4. Results
Revise Table 3 to comply with the conference formatting guidelines.
The research results are structured appropriately to address the study's objectives.
The confirmatory factor analysis, model fit of the revised model, and reliability are all well organized.
5. Discussion
The discussion section is well-structured.
However, it appears to contain more explanations of the research results than the intended focus of the study.
It would enhance the paper to explain the applicability in practice based on the results, which confirm validity and reliability.
Section 4.1, Strengths and Limitations, is well-organized.
Furthermore, the suggestions for future research are excellent and could be moved to the conclusion section.
6. Conclusion
The suggestions for future research are highly commendable.
7. Reference
Revise the references to align with the formatting guidelines of the conference.
Sentences written in a compound sentence do not seem to convey the meaning well.
I hope that the researcher's argument can be conveyed clearly by writing in short sentences.
Author Response
I hope this letter finds you well. I had the opportunity to review your article titled, “Adaptation and psychometric properties of the Behavioral Regulation in Exercise Questionnaire (BREQ-3) for motivation towards incidental physical activity”, which was submitted Behavioral Sciences.
Comment 1: Abstract. Please provide a brief explanation of the research methods.
Response 1: In the abstract of the manuscript, specifically between lines 26 and 32, we have provided the main information about the methodology used in the study. We begin by mentioning the instrumental design of the study since it corresponds to the analysis of the psychometric properties of an adaptation of a self-report instrument. In addition, the sample studied (n=346, university students in Chile) and its main characteristics (mean age of 21.1 ± 2.6 years, 61.3% women) are described. The statistical analyses used are also detailed, such as exploratory factor analysis and internal consistency calculations using Omega and Alpha coefficients, among others.
Comment 2: Introduction. When presenting previous studies in the text, place citations at the end of sentences.
Response 2: We are grateful for the reviewer's comment, which allowed us to improve the writing of the manuscript. In this sense, we have made some modifications, placing the quotations at the end of sentences. This also helped us to shorten the length of the sentences, thus making them easier to understand.
Now, between lines 45 to 48, it says: The benefits of increasing physical activity include improved health and well-being (Dempsey et al., 2021; Díaz-Martínez et al., 2018; Kraus et al., 2019; Lacombe et al., 2019; Zhao et al., 2020). So promoting physical activity is a public health priority (Stanaway et al., 2018; World Health Organization, 2018).
Also, between lines 93 to 101, it says: In its first version, it opted for a 15-item model grouped into four factors. With adequate model fit indices (χ² = 277.19, p < .001, Root Mean Square Error of Approximation or RMSEA = .07) and internal consistency using Cronbach's alpha (external regulation = 0.79, introjected regulation = .78, identified regulation = .79, intrinsic motivation = .90). This first version did not consider the dimension of amotivation because it was statistically problematic and left out integrated regulation (Mullan et al., 1997). In this way, the model consisted of the dimensions of external regulation, introjected regulation, identified regulation, and intrinsic motivation (Mullan et al., 1997).
However, we have kept other quotes in their original location, as changing their position could weaken their connection to the information they support. In addition, placing them at the end together with other quotations could generate confusion, giving the impression that arguments that do not correspond to the content of these references are being supported.
Comment 3: Each paragraph should consist of at least three sentences. The verbs in the theoretical background should be written in the past tense. There is a tendency for sentences to become lengthy, leading to ambiguity. Write concise sentences to ensure the author's arguments are conveyed accurately.
Response 3: Based on the reviewer's comment, we have reviewed the wording of the manuscript and the verb tenses used in the background again. All the co-authors reviewed the wording jointly and subsequently by a native English translator. The modifications made, both in writing and in response to the other comments of the reviewers, have been recorded in the manuscript file with change control, which we have uploaded to the platform.
This study's purpose of exploring the applicability of BREQ is considered excellent. In particular, survey instruments need continuous validation and reliability testing over time. From this perspective, this study is highly interesting and necessary in the field.
Comment 4: Materials and Methods [63] Adjust the font size appropriately.
Response 4: We appreciate the reviewer's comment. We have fixed the citation formatting error and adjusted the font size according to the journal's guidelines.
Comment 5: 2.2Participants. The study participants consist solely of women. If so, the title should specify that the study is limited to women. Additionally, explain why the study focuses specifically on college students.
Response 5: In relation to the participants, these were university students of both sexes. As detailed in the corresponding section, 61% of the sample corresponded to women. On the other hand, university students were selected as the study population for several reasons, mainly methodological. First, it is a population with access and feasibility to reach the sample sizes required for the analyses used in this type of study. In addition, university students present homogeneity in sociodemographic characteristics such as age, educational level, and social context, which reduces external variability and facilitates the specific evaluation of the instrument's properties. Despite this homogeneity, there is also diversity in aspects such as gender, academic careers, and cultural background, which allows the instrument to be evaluated in a varied but controlled group. Finally, validating an instrument in university students constitutes an initial step towards broader studies in other populations, serving as a preliminary frame of reference.
Comment 6: 2.3 Instruments and 2.4 Procedure. These sections are very well written. Ensure uniformity in all mathematical symbols.
Response 6: We have carefully reviewed these sections to ensure uniformity in all mathematical symbols used.
Comment 7: 4. Results. Revise Table 3 to comply with the conference formatting guidelines.
Response 7: We have modified Table 4 (formerly Table 3) to fit the format required by the journal.
The research results are structured appropriately to address the study's objectives. The confirmatory factor analysis, model fit of the revised model, and reliability are all well organized.
Comment 8: 5. Discussion. The discussion section is well-structured. However, it appears to contain more explanations of the research results than the intended focus of the study.
It would enhance the paper to explain the applicability in practice based on the results, which confirm validity and reliability.
Response 8: Thank you for your feedback on the discussion section. We appreciate your suggestion to highlight the applicability in practice based on the results. This aspect was addressed between lines 406 and 417, where we discuss how the adaptation of the BREQ-3 contributes to research on motivation towards incidental physical activity. Specifically, we emphasize that evidence on the psychological mechanisms explaining these behaviors remains limited (Levine, 2007; Reynolds et al., 2014; Tudor-Locke et al., 2007). Current findings focus on the interaction between the continuum of motivation and implicit processes, such as attitudes, attentional biases, and approach-avoidance mechanisms (Oliver & Kemps, 2018). Furthermore, we note how integrating the self-determination continuum into the Theory of Planned Behavior can enhance understanding of physical activity behaviors (Gómez-Mazorra et al., 2022; M. S. Hagger & Chatzisarantis, 2014) and how self-determined motivations mediate between personality traits, constructs of the Theory of Planned Behavior, and physical activity (Kekäläinen et al., 2022).
Comment 9: Section 4.1, Strengths and Limitations, is well-organized. Furthermore, the suggestions for future research are excellent and could be moved to the conclusion section.
Response 9: We agree with the reviewer that this information should be transferred to the conclusions. Therefore, between lines 424 and 428, we have made some changes to the wording to reflect this change. The text now reads as follows: Future research should address the limitations of this research, such as sample size and lack of measures of concurrent validity and temporal stability. However, this adaptation of the BREQ-3 offers a valid and reliable tool in the study of the psychological mechanisms underlying motivation towards incidental physical activity, an area that has so far received little attention.
Conclusion. The suggestions for future research are highly commendable.
Comment 10: Reference. Revise the references to align with the formatting guidelines of the conference.
Response 10: We have adjusted the citation and reference format to the new format established by the journal, which corresponds to the APA style.
Comment 11: Comments on the Quality of English Language. Sentences written in a compound sentence do not seem to convey the meaning well. I hope that the researcher's argument can be conveyed clearly by writing in short sentences.
Response 11: We appreciate the reviewer's comment, which allowed us to improve the quality of writing of our manuscript and, with it, the clarity of the ideas we wish to convey. To address this suggestion, the co-authors have carefully reviewed the text, in addition to reviewing a native English translator and using Grammarly software. As a result, some sentences have been shortened to improve the wording and make it easier to understand. All modifications have been recorded in the manuscript using the change control.